# Mental Health Recovery in Social Psychiatric Policies: A Reflexive Thematic Analysis

**DOI:** 10.3390/ijerph20126094

**Published:** 2023-06-09

**Authors:** Jacob M. Nielsen, Niels Buus, Lene L. Berring

**Affiliations:** 1Psychiatric Research Unit, Centre for Relation & De-Escalation, Mental Health Services, 4200 Slagelse, Denmark; lelb@regionsjaelland.dk; 2Monash Nursing and Midwifery, Faculty of Medicine, Nursing and Health Sciences, Monash University, Clayton 3800, Australia; niels.buus@monash.edu; 3Department of Regional Health Research, University of Southern Denmark, 5230 Odense, Denmark

**Keywords:** recovery, mental health, management, policy, knowledge bases, social psychiatry, discourses, governmentality

## Abstract

The realisation of recovery as an overarching goal of mental health care services has proven difficult to achieve in practice. At present, concepts of recovery are contested and unclear, which affects their implementation in psychiatric practices. We examined social psychiatric policies about recovery with the aim to explore their underlying assumptions about recovery. Relevant texts from the policies’ knowledge bases were subjected to reflexive thematic analysis. We developed a central theme: “A clinical standardisation of the concept of recovery”. The theme involved meaning clusters that encompassed conflicting and commonly shared assumptions about recovery across the text corpus. We discussed the findings from discourse analytical and governmentality perspectives. In conclusion, the policies’ aim of providing clarity about recovery was circumvented by the very knowledge bases used to support their endeavours.

## 1. Introduction

Internationally, and in Anglophone countries in particular, governments have struggled to implement recovery policies as an overarching goal of mental health care services [1,2]. However, the realisation of these ambitions has been permeated by difficulties and contestation concerning the use of recovery in practice [3,4,5]. The contestation and the challenges surrounding recovery are not isolated to a singular domain of psychiatric practice but have permeated mental health care across settings and sectors. Despite this, processes of implementation have continued, which in turn have accelerated concerns for the future of the concepts and practices of recovery [6]. Therefore, it is relevant to study health care policies on recovery. In the current paper, we will analyse the knowledge bases of two policies from the Danish National Board of Social Services providing guidance about recovery in social psychiatry. A policy’s knowledge base is the literature that it refers to and constitutes the empirical and theoretical underpinnings of the policy.

This study contributes to ongoing discussions about social psychiatric health policies and the barriers that have challenged the implementation of recovery in mental health care services. It is an examination of the dynamics at play in the translation of mental health aspirations into policy.

### 1.1. Theoretical Understandings of Recovery

Recovery in mental health involves several distinct types or conceptions [7]. The following is a presentation of three theoretical understandings of recovery relevant to the present study: (1) clinical, (2) personal, and (3) social conceptions of recovery. A description of the three types of recovery will be unfolded next.

**Clinical recovery** is understood as full symptom remission from mental illness, thereby regaining the function level that existed before the illness set in and maintaining prolonged employment in a competitive labour market [8]. Clinical recovery relies on certain objective criteria of remission from mental illness, which is rooted in a psychiatric dichotomy between ‘normality’ and ‘illness’ [9]. Clinical recovery is inherently agnostic about economic and social structures. **Personal recovery** is understood as the personal and unique process towards a satisfying and contributing life even with the limitations caused by mental illness [10,11,12,13]. The understanding of personal recovery is permeated by ambiguity and complexity [11,12,13], and it has been attributed different connotations over time [7]. This understanding of recovery originated during a period of social upheaval and the transformation of public organisations and institutions (the 1960s). The transformations involved the restructuring of the services provided by a variety of organisations such as psychiatric hospitals (or asylums given the historical context), facilities for persons with handicaps, and prison complexes. Against this backdrop of social transformation and the restructuring of established service provisions, the research into recovery conducted in the 1970s and 1980s highlighted the importance of the social aspects of recovery; i.e., recovery was conceived as a personal process within a social context. However, the later conceptions of recovery from the 1990s and onwards have generally disregarded the structural and environmental circumstances as essential for the realisation of personal recovery [7]. **Social recovery** is understood as the realisation of residential and economic independence in combination with low social disruption, which means the upholding and development of social networks as well as individual social relationships [14,15]. In the case of social recovery, partial remission can be actualised despite the presence of residual symptoms of mental illness [13]. The realisation of social recovery also involves a high level of functionality, which makes full-time or at least part-time employment possible in a competitive labour market. Generally, the need for continuous psychiatric treatment is low or non-existing, and medication doses are well below the levels of average medication regimes. Social recovery incorporates certain aspects of clinical recovery, i.e., economic self-sufficiency and a high level of functionality, especially in an employment perspective with some common denominators of personal recovery [14].

### 1.2. Theoretical Perspective

The different understandings of the concept of recovery and the contestation permeating its use in psychiatric practices raise questions about the concept’s future: The realisation and successful implementation of programmes, procedures, and goals depends on the conditions of the possibilities of an organisation [16,17]. Effectively, social psychiatric policy is developed against a backdrop of psychiatric discourses, which include and exclude certain types of knowledge from the process of development. Thereby, an operational policy and its successful implementation depend on and reflect the possibilities of a psychiatric organisation. We will be using a Foucauldian perspective on discourses in this study, which concerns the discursive construction of conditions of possibilities in psychiatric practices [16,17].

We will also be using a Foucauldian perspective on governmentality. This perspective concerns the use of a singular management rationale in liberal societies. It is defined as *the conduct of conduct* and aims to develop potentiality via intervention among its target group. The development of potentiality concerns the optimisation of individual capacities and personal autonomy, i.e., the production of autonomous persons capable of self-management. Governmentality thereby exhibits the productivity of power in its development of potentiality [18]. In a governmentality perspective, the National Board of Social Services’ policies can be seen as managerial instruments on a micro level in terms of guiding the municipalities’ management of social psychiatry, i.e., the conduct of conduct. For instance, outpatient treatment programmes depend on the patients’ realisation of autonomy, accountability, and self-conduct to function adequately, whereby the productivity of power is used in social psychiatric practices to achieve optimisation and institutional functionality. The governmental articulation of certain recovery practices creates a semantic shift in the use of interventions. This shift negates the traditional distinction between deficit and optimization [18]. The development of potentiality constitutes the focal point of intervention whereby social circumstances are effectively discarded as being inconsequential. The use of this perspective on recovery will support our reading of the analysis.

### 1.3. Aim

Considering the contested meaning of recovery, the aim of this paper is to explore the assumptions about recovery in the knowledge bases of two policies from the National Board of Social Services. This will provide insight into the contested meanings of recovery and contribute to the prospective use of recovery.

## 2. Study Design

This study was based on a qualitative approach with a collection of texts subjected to reflexive thematic analysis [19]. This approach was applied to achieve the paper’s aim, which required oversight over the text corpus in an analytical sense. A reflexive thematic analysis was used because of its inherent strengths in processing large quantities of data. The reflexive thematic analysis was also compatible with the study’s theoretical perspective: the meaning clusters and theme could be viewed as analytically produced manifestations of the discursive construction of conditions of possibilities concerning recovery across the text corpus. The compatibility of the study’s analysis method and theoretical perspective potentiated the investigation of the policies’ knowledge bases.

### 2.1. Text Sample and Policy Selection

The texts were sampled among the references listed in the National Board of Social Services’ policies for the Danish municipalities: “Professional management of a recovery-oriented rehabilitating practice; Seven guiding principles for the professional management of social psychiatry” [20] (our translation) (Policy A) and “People with mental health challenges; Effective social measures” [21] (our translation) (Policy B). We included texts, listed in the policies’ reference list, using the selection criterion ‘texts concerning recovery’. The final sample included 23 texts (813 pages).

Policy A was published in 2021 [20] and is one of the most recent contributions to a growing corpus of policies about recovery in a Danish context. Policy B was published in 2013 [21] and is one of the first national policies about recovery in Denmark. The two policies were closely related because of their institutional anchoring and shared a focus on managerial aspects of the implementation of recovery in social psychiatry [20,21]. These aspects were decisive for the selection of the policies and supported the exploration of the assumptions about recovery in the policies’ knowledge bases.

### 2.2. Context

The mental health sector in Denmark was organised at two levels constituted by (1) regions and (2) municipalities. Each level had distinct obligations and responsibilities concerning their respective service provisions. The regions provided clinically founded inpatient treatment, whereas the municipalities delivered outpatient treatment and supportive housing in the context of civil society. The National Board of Social Services developed and provided national guidance by policies based on the best available knowledge for the municipalities’ social psychiatric services. The National Board of Social Services’ policies were not legally binding documents.

Policy A concerned professional management in social psychiatric organisations in support of personal recovery. The policy’s aim was to increase clarity about and awareness of the direction of the organisation as well as its practices among middle managers and staffers alike. The policy was aimed at higher level management to effect managerial change throughout social psychiatric organisations [20]. Policy A was conducted and facilitated by the consulting firm “Implement Consulting Group” in cooperation with the National Board of Social Services. The consulting firm facilitated implementation at large in both public and private organisations [22]. There was no mention of how the cooperation between the two parties affected the development of the policy [20].

Policy B concerned the development of middle management in support of social recovery across social psychiatric settings. In particular, the policy was aimed at middle managers, professional coordinators, and development consultants involved in the planning and development of social psychiatric practices in Danish municipalities. The policy highlighted specific practices and methods in support of social recovery focusing on their effectiveness as well as descriptions of different implementation processes. Policy B was conducted and facilitated by a team under the National Board of Social Services with in-depth knowledge about mental health care and recovery. There was no mention of specific contributions from individual team members regarding the development of the policy [21].

### 2.3. Reflexive Thematic Analysis

The reflexive thematic analysis [23] involved six distinct but interconnected steps:

(1) Familiarising ourselves with the texts, which meant reading and re-reading them. We identified the parts of the 23 texts that explicitly or implicitly were concerned with recovery and began developing initial codes.

(2) Initiation of the coding procedure, i.e., the coding of relevant features in the texts. An example of the coding procedure is provided by this extract from a text referenced by Policy B: “Clinical recovery is an idea originating from psychiatric and social psychiatric professionals, which has its basis in a reduction of symptoms, regaining social functionality and other ways of ‘returning to normality’.” [24] (our translation). This was coded as “A clinical focus on regaining ‘normality’.” The coding process produced 119 individual codes.

(3) The individual codes were then organised around central ideas or concepts. In this process of organising codes, we considered the interconnectedness between them.

(4) The codes were reread to ensure that the theme satisfactorily encompassed the coded data. Furthermore, this step of the analytical process involved the development of thematic maps, i.e., the production of meaning clusters. The meaning clusters were centred around *methodological pluralism*, *multiplicity of definitions*, *risk reduction*, and *clinical dominance*, which were viewed as analytically produced manifestations of the discursive construction of conditions of possibilities concerning recovery across the text corpus.

(5) The definition of the theme and the development of a title central to it. This meant an ongoing refinement of the theme as well as the development of a clear title, which led to the production of the following theme: *A clinical standardisation of the concept of recovery*. The connection between theme and meaning clusters was one of attribution in which the theme incorporated the meaning of central ideas, cf. the above-mentioned discursive construction of conditions of possibilities.

(6) In the final step of the analytical process, we incorporated extracts from the data set before a discussion of the analysis in relation to the research question; this was initiated in a reflexive manner. Lastly, we wrote up the analysis, which contained the presentation of the analytical findings as well as a discussion of these in relation to the research question.

## 3. Results

### 3.1. Introduction to the Text Corpus

The texts had different characteristics depending on the objectives of the policies. The texts in Policy A were relatively homogeneous. They were predominantly constituted by theory about recovery and, to a lesser degree, incorporated empirically based research. The texts encompassed theoretically based policies and management reports providing guidance for practice (see, for instance, [25,26]) as well as papers concerning the social psychiatric practice that included some empirical founded data (see [27,28]). All the texts were produced in and focused on circumstances concerning the particular Danish context.

The texts in Policy B were relatively heterogeneous, with most texts reporting empirical analyses. A few texts used theoretical–analytical investigations of recovery, see for instance [3,29,30,31]. The empirically based research was predominantly from North America and tended to focus on interventions and treatment programmes (see [32,33,34]), whereas the knowledge generated in Scandinavia tended to focus on the current state of the mental health services and the prospective scenarios of development in relation hereto (see [35,36]). These latter texts were primarily conducted and facilitated by professionals on behalf of health and policy advisory institutions, including, among others, the Danish Health Authority and its Scandinavian counterparts.

### 3.2. Findings from the Reflexive Thematic Analysis: A Clinical Standardisation of the Concept of Recovery

We developed a theme: “A clinical standardisation of the concept of recovery”. The theme was defined as the elevation of a clinical understanding of and approach to the practice of recovery at the expense of alternative conceptions of recovery in psychiatry. This conceptualisation of recovery dominated as it was easily compatible and could be integrated into the established power structures of the psychiatric services and organisation. The theme involved meaning clusters about methodological pluralism, the multiplicity of definitions, clinical dominance, and risk reduction in psychiatry, which encompassed conflicting and commonly shared assumptions about recovery across the text corpus.

### 3.3. Theme: A Clinical Standardisation of the Concept of Recovery

Firstly, methodological pluralism was apparent, which meant that interventions, treatment programmes, and management were used indiscriminately across several texts [24,25,26,36,37,38]. Methodological pluralism contributed to a circumvention of consensus concerning its use, which gave preference to a standardised clinical conceptualisation of recovery. The compatibility and integrability of a clinical conceptualisation of recovery with the discursive power structures in psychiatry potentiated the dominance of clinical recovery on behalf of alternatives hereto. The alternative conceptions of recovery were at odds with the established structures of power and the prevailing practices in psychiatry.

Secondly, there was a multiplicity of definitions of recovery across the texts [26,28,39]. The following data example is extracted from a text referenced by Policy B and encapsulates an indiscriminate generic definition of recovery: “Essentially, recovery means that there is hope for the individual to gain full or partial recovery from even severe mental illness. Recovery is achieved by the patient’s active participation herein.” [40] (our translation). This example illustrates a generic definition of recovery. Because of its lack of specification, it achieves universal validity at the expense of clarity about recovery. Similar definitions and conceptualisations of recovery were widely distributed throughout the texts about psychiatric practices (see, for example [27,28]) as well as most of the texts involving research about recovery (see, for example [30,41]). Recovery appeared as a one-size-fits-all concept inherently favouring the clinical epistemology of recovery over alternative conceptions. The implementation of procedures, programmes, and practices aligned with the established practices excluded alternative understandings of recovery.

Finally, risk reduction and clinical dominance was a manifestation of the psychiatric approach to and understanding of mental illness and recovery, which was apparent across several texts [24,35,36]. The use of risk reduction and clinical dominance in psychiatry was rooted in governmentality as well as psychiatric discourses about symptoms of mental illness. This extract from the text corpus encompassed the twofold aim of risk management and optimisation (governmentality): “The best way to reduce the likelihood of crises from occurring is to develop skills of self-management. These skills create actor awareness, empowerment, and resilience to manage relapse. The ability to recognise and respond to symptoms of mental problems is an important skill. The challenge in relation to recovery is to work with early warning signals in a way that strengthens people’s ability to self-regulate instead of creating anxiety about and unnecessary attention to potential relapses.” [24] (our translation). It accentuated the connection between optimisation (governmentality) and prophylactic practices by potentiating the patients’ capabilities of self-monitoring and self-governance. Risk reducing measures were deemed legitimate if certain clinical criteria were met, i.e., if the mental state of a patient was considered to pose a risk for themselves or others by the mental health professionals. This legitimised and potentiated the use of coercion if the patient did not “collaborate” voluntarily [24,35,36]. This potentiated the reversal of certain fundamentals of recovery; e.g., a temporary violation of someone’s bodily integrity as a way of responding and reducing risk could be read as a negation of individual autonomy and empowerment. In this perspective, the dignity of risk was circumvented by the elevation of clinical considerations over alternative (non-clinical) interpretations of and approaches to risk. The combination of practical necessity and the clinical rationalisation of the risk reduction efforts produced a singular understanding of the phenomenon of risk that was deemed positive for the process of recovery in a clinical sense. The clinical determination of risk was founded on paternalistic notions that simplified an inherently multifaceted and complex matter.

## 4. Discussion

### 4.1. The Assumptions about Recovery

The findings revealed a lack of consensus about recovery regarding methodology, definitions of recovery, and discrimination between different types of recovery. Despite this, certain commonly shared assumptions about recovery were identified across the text corpus: they constituted clinical concerns and risk management, which, informed by psychiatric discourses, effectuated a clinical standardisation of the concept of recovery in psychiatric practices. Essentially, a clinical standardisation of the concept of recovery was a manifestation of the conditions of possibilities concerning recovery. For instance, a clinical understanding of and approach to recovery and mental illness informed the psychiatric conduction of risk management, which dictated the psychiatric response to expressions of risk (based on a clinical interpretation of risk). This could affect patient autonomy negatively and potentially effectuate an increased application of coercive measures in psychiatry. In effect, the standardisation of the concept of recovery designated room for manoeuvre for mental health professionals and psychiatric patients in accordance with the limitations and possibilities of standardisation.

### 4.2. A Clinical Standardisation of the Concept of Recovery

These findings about a clinical standardisation of the concept of recovery were supported by similar findings in established Danish recovery research [1,4,5]. These studies applied a practice-oriented approach to recovery [4,5] and a discourse analytical investigation of the practice of recovery [1]. In contrast, this study used a theoretical–analytical perspective to investigate texts referenced by two policies about recovery. The current study highlighted how the applied biomedical treatment practices supported a clinical understanding of recovery. This approach to and use of recovery potentiated an exclusive focus on pharmacologically founded treatment regimens, whereby the use of recovery was permeated by an exclusively clinical understanding of treatment [1,4,5]. The current study also highlighted the standardisation of treatment options as characteristic of this (clinical treatment supported recovery). This practice subjected patients to a one-size-fits-all approach to treatment-supported recovery and simultaneously relegated the responsibility for the actualisation of clinical recovery to the individual patient. In effect, a “take it or leave it scenario” emerged on the grounds of the standardisation of clinical treatment [1,12]. The findings in the established research corresponded to the main finding of this study: a clinical standardisation of the concept of recovery. For instance, the combination of biomedical treatment regiments with a standardisation of treatment practices [1] excluded non-clinical conceptions of recovery in the psychiatric practice [4,5]. Despite the different approaches deployed by the above-mentioned studies in the investigation of the phenomenon, the similarity of the findings was a quite remarkable indication of the inadequacies that permeate the current state of recovery.

The use of risk reduction efforts underlined the interventional nature of psychiatry. The articulation of the psychiatric response to the question of risk was dictated by governmentality, which incorporated practical necessity (for the actualisation of optimisation) and reductions in negative expressions of risk. Against this backdrop of governmentality, the use of clinical dominance and risk reduction efforts discarded alternative understandings of and approaches to recovery from gaining access to the established psychiatric practices. Intervention was used to potentiate optimisation, i.e., a manifestation of the productivity of power [18]. The manifestation of this management rationale (governmentality) in psychiatric treatment practices highlighted the importance of compatibility between management rationales and treatment ideals for the social practice of the organisation. Thereby, it seemed as if a clinical standardisation of the concept of recovery was the most compatible and effectively the most useful conception of recovery in the psychiatric practices under the prevailing circumstances. The fact that the implementation of recovery in psychiatry was politically dictated underlined the fundamental connection between governmentality and the management of recovery in psychiatric organisations [1].

The findings about a clinical standardisation of the concept of recovery conflicted with the aim of the policies to provide clarity about recovery in social psychiatry. For instance, the policies differentiated between different understandings of recovery in their use of the concept. They also highlighted the importance of consensus and stringency in the understanding of and approaches to recovery. Despite the policies’ aims and intentions about these aspects of recovery, the present analysis identified an inadequate discrimination between different types of recovery. This was further complicated by a lack of meaningful definitions of recovery, which diluted its conceptual meaning. Additionally, the present study revealed a methodological pluralism, which involved an indiscriminate use of interventions, treatment programmes, and management practices that circumvented the consensus and stringency of the knowledge bases. Effectively, the assumptions about recovery in the knowledge bases conflicted with the aims of the policies on fundamental areas of recovery, which circumvented the usefulness of the policies because of manifest incongruence between the two parts.

### 4.3. Strengths and Limitations

The description of the epistemological position and the implementation of the reflexive analytical method provided transparency about the execution of this study. The transparency supported the internal validity and reliability of the study by consistently detailing the processes involved in its execution. The involvement of several researchers in the performance of the study also increased its internal validity based on their discussions about research findings and their cooperation in the development of a theme. The theme was developed by thorough discussions about meaning, whereby the researchers reached a consensus. However, the research findings could potentially be different from the ones presented if a different research perspective, sample strategy, or analysis method had been used in the execution of the study. However, similar reservations would apply to constructivist research in general.

## 5. Conclusions

A more nuanced understanding of the concept of recovery and a clarification of its contested meaning is essential for the future of recovery in management, practice, and research alike. Stronger congruence between policies and their knowledge bases should be pursued to reduce uncertainty about the potential meanings of recovery and their use in practice. An increased awareness of the differences between the types of recovery available could support the utilisation of alternative understandings of recovery in psychiatric practices. Finally, a thorough discrimination between different types of recovery could reduce a dilution of some of the non-clinical meanings of recovery.

## Data Availability

No new data were created or analysed in this study. Data sharing is not applicable to this article.

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
