# Peer review of "Mental Health Recovery in Social Psychiatric Policies: A Reflexive Thematic Analysis"

_ijerph, 2023, doi:10.3390/ijerph20126094_

Round 1

Reviewer 1 Report

dear Authors

Your paper Mental health recovery in social psychiatric policies: A reflexive thematic analysis is very interesting and important. Foucauldian theories are very illuminating in this study. You have done good work and reported it well. I got only some minor comments

1) What is the position of social psychiatry in regards to academic and clinical mental health practices or in terms of  government and planning of mental health policy and practice in Denmark?

2) What is the role of the National Board of Social Services in Denmark in formation and government of mental health policy and practice  in Denmark? How mental health policy is formed in Denmark, who are the institutions in charge? How mental health practices are followed in national level?

3) You analysed relevant texts from the policies’ knowledge bases. Why you selected these two papers? Are there more governmental instructions of mental health policy in Denmark? What is the relationship of these two analysed papers to each other? What is the autonomy of municipalities in organising mental health services?

5) are the public discussion going on about alternative recovery practices in mental health care? By patient organisations for example? Since using only clinical recovery concept is little bit conservative and old fashioned today.

Author Response

Response to reviewers

Thank you for carefully reading the manuscript. We have made good use of most suggestions and reflections and pushed back on a few where we argue that the suggested changes do not quite fit the paper’s plot. We have highlighted changes in the text with red colour.

Reviewer #1

1) What is the position of social psychiatry in regards to academic and clinical mental health practices or in terms of government and planning of mental health policy and practice in Denmark?

2) What is the role of the National Board of Social Services in Denmark in formation and government of mental health policy and practice in Denmark? How mental health policy is formed in Denmark, who are the institutions in charge? How mental health practices are followed in national level?

Our response (1 and 2): Thank you for making us aware of this need. We have followed your advice and added the following to “Context”:

“The mental health sector in Denmark was organized at two levels constituted by 1) regions and 2) municipalities. Each level had distinct obligations and responsibilities concerning their respective service provisions. The regions provided clinically founded inpatient treatment whereas the municipalities delivered outpatient treatment and supportive housing in the context of civil society. The National Board of Social Services developed and provided national guidance by policies based on best available knowledge for the municipalities’ social psychiatric services. The National Board of Social Services’ policies were not legally binding documents.”

3) You analysed relevant texts from the policies’ knowledge bases. Why you selected these two papers? Are there more governmental instructions of mental health policy in Denmark? What is the relationship of these two analysed papers to each other? What is the autonomy of municipalities in organising mental health services?

Our response (3): We have added the following to “Text sample and policy selection”:

”Policy A was published in 2021 [20] and one of the most recent contributions to a growing corpus of policies about recovery in a Danish context. Policy B was published in 2013 [21] and one of the first national policies about recovery in Denmark. The two policies were closely related because of their institutional anchoring and shared focus on managerial aspects of the implementation of recovery in social psychiatry [20,21]. These aspects were decisive for the selection of the policies and supported the exploration of the assumptions about recovery in the policies’ knowledge bases.”

Several social psychiatric policies about recovery have been published by the National Board of Social Services in addition to the ones included in the study. However, we determined that including additional social psychiatric policies about recovery from the National Board of Social Services was redundant with the study’s scope in mind. As stated above, we have added a brief explanation about the Danish municipalities’ roles to the section: “Context”.

4) Are the public discussion going on about alternative recovery practices in mental health care? By patient organisations for example? Since using only clinical recovery concept is little bit conservative and old fashioned today.

Our response (4): Discussions about recovery are ongoing, but it’s largely occurring outside of established mental health care sectors by academics, researchers, patient organisations, and activists. In effect, the policies’ knowledge bases do indeed reflect an old fashioned and quite conservative approach to recovery, cf. the study’s findings and discussion of the results. Unfortunately, this doesn’t leave much room for manoeuvre for alternative understandings of and approaches to recovery in discussions about recovery in Denmark (including the policies’ knowledge bases).

Reviewer 2 Report

The authors of this paper employ the guidelines of reflexive thematic analysis in the examination of two Danish National Board of Social Services policy documents to explore the concept of recovery in the context of social psychiatric policy. They outline the steps followed in the analyses and provide findings leading to their conclusion that there is a fundamental discrepancy between the stated recovery policy and the supportive published knowledge bases. They argue for greater awareness of the types of recovery, noting that current conceptualizations undermine effective realization of the broad goals of a recovery model.

Author Response

Response to reviewers

Thank you for carefully reading the manuscript. We have made good use of most suggestions and reflections and pushed back on a few where we argue that the suggested changes do not quite fit the paper’s plot. We have highlighted changes in the text with red colour.

Reviewer #2

Our Response: Thank you so much. We are thankful for the positive feedback you have provided.

Reviewer 3 Report

1. The title focuses on analyzing social psychiatric policies. It would be beneficial for the author to include additional information regarding the development, viewpoints, and literature review of these policies.

2. In the section discussing theoretical perspectives, it would be helpful for the author to provide more information on how policy reflects critical viewpoints from a Foucauldian perspective.

3. When describing the study design, it would be advantageous for the author to offer more theoretical discourse about why reflexive thematic analysis was chosen as a method and its importance in this context.

4. In the discussion section, it is recommended that the author provides further details about both theoretical values and practical suggestions related to exploring policy development and reflections.

Author Response

Reviewer #3

1) The title focuses on analysing social psychiatric policies. It would be beneficial for the author to include additional information regarding the development, viewpoints, and literature review of these policies.

Our response (1): We have added the following to “Context”:

“The mental health sector in Denmark was organized at two levels constituted by 1) regions and 2) municipalities. Each level had distinct obligations and responsibilities concerning their respective service provisions. The regions provided clinically founded inpatient treatment whereas the municipalities delivered outpatient treatment and supportive housing in the context of civil society. The National Board of Social Services developed and provided national guidance by policies based on best available knowledge for the municipalities’ social psychiatric services. The National Board of Social Services’ policies were not legally binding documents.”

and “Text sample and policy selection”:

”Policy A was published in 2021 [20] and one of the most recent contributions to a growing corpus of policies about recovery in a Danish context. Policy B was published in 2013 [21] and one of the first national policies about recovery in Denmark. The two policies were closely related because of their institutional anchoring and shared focus on managerial aspects of the implementation of recovery in social psychiatry [20,21]. These aspects were decisive for the selection of the policies and supported the exploration of the assumptions about recovery in the policies’ knowledge bases.”

Thank you for making us aware of this need. We hope the additions above makes the text clearer.

2) In the section discussing theoretical perspectives, it would be helpful for the author to provide more information on how policy reflects critical viewpoints from a Foucauldian perspective.

Our response (2): We have added the following to “Theoretical perspective”:

“Essentially, policy is developed against a backdrop of psychiatric discourses, which include and exclude certain types of knowledge, whereby an operational policy and its successful implementation depend on and reflect the discursive possibilities in an organisation.”

and

“In a governmentality perspective, the National Board of Social Services' policies can be seen as managerial instruments on a micro level in terms of guiding the municipalities’ conduction of social psychiatry, i.e., the conduct of conduct. For instance, outpatient treatment programmes depend on the patients’ realisation of autonomy, accountability, and self-conduct to function adequately, whereby the productivity of power is used in social psychiatric practices to achieve optimisation and institutional functionality.”

3) When describing the study design, it would be advantageous for the author to offer more theoretical discourse about why reflexive thematic analysis was chosen as a method and its importance in this context.

4) In the discussion section, it is recommended that the author provides further details about both theoretical values and practical suggestions related to exploring policy development and reflections.

Our response (3 and 4): Thank you for highlighting these aspects. We have added the following to “Study design” and “Theme: A clinical standardisation of the concept of recovery”. The latter has been added to the analysis and not the discussion as suggested. Despite the caveat, we believe the addition unfolds theory in the exploration of recovery. We hope these additions makes the text clearer.

”This approach was applied to achieve the paper’s aim, which required oversight over the text corpus in an analytical sense. A reflexive thematic analysis was used because of its inherent strengths in processing large quantities of data. The reflexive thematic analysis was also compatible with the study’s theoretical perspective: The meaning clusters and theme could be viewed as analytically produced manifestations of the discursive construction of conditions of possibilities concerning recovery across the text corpus. The compatibility of the study’s analysis method and theoretical perspective potentiated the investigation of the policies’ knowledge bases.”

and

“This extract from the text corpus encompassed the twofold aim of risk management and optimisation (governmentality): “The best way to reduce the likelihood of crises from occurring is to develop skills of self-management. These skills create actor awareness, empowerment, and resilience to manage relapse. The ability to recognise and respond to symptoms of mental problems is an important skill. The challenge in relation to recovery is to work with early warning signals in a way that strengthens people's ability to self-regulate instead of creating anxiety about and unnecessary attention to potential relapses.” [24] (Our translation). It accentuated the connection between optimisation (governmentality) and prophylactic practices by potentiating the patients' capabilities of self-monitoring and self-governance.”

Reviewer 4 Report

  1. The analysis could be improved by clarifying the relationship between the two theoretical perspectives under consideration. Providing a more explicit link between the Foucauldian perspectives on discourses and governmentality will help readers better understand their relationship.
  2. The review would be more effective if it included specific examples or further elaboration on the theoretical concepts mentioned, such as governmentality. Providing brief examples or explanations will enhance the accessibility of these concepts in the context of psychiatric recovery practices.
  3. In line 231, the statement "effectuated a clinical standardization of the concept of recovery" could be further explained to help readers understand its meaning, addressing how clinical concerns and risk management techniques led to standardization and the implications of this standardization.
  4. More references should be included in the article, especially in lines 45-74, where various types of recovery are mentioned but only one reference is provided for each type.
  5. The transformation in personal recovery, discussed in lines 56-61, is unclear and requires further explanation. To facilitate reader comprehension, the time scales for each transformation should be better organized.
  6. In section 1.1, A table could be added to compare the theories 

Author Response

Response to reviewers

Thank you for carefully reading the manuscript. We have made good use of most suggestions and reflections and pushed back on a few where we argue that the suggested changes do not quite fit the paper’s plot. We have highlighted changes in the text with red colour.

Reviewer #4

1) The analysis could be improved by clarifying the relationship between the two theoretical perspectives under consideration. Providing a more explicit link between the Foucauldian perspectives on discourses and governmentality will help readers better understand their relationship.

Our response (1): We have added the following to “Reflexive thematic analysis”:

”, which were viewed as analytically produced manifestations of the discursive construction of conditions of possibilities concerning recovery across the text corpus.”

and

”, cf. the above-mentioned discursive construction of conditions of possibilities.”

2) The review would be more effective if it included specific examples or further elaboration on the theoretical concepts mentioned, such as governmentality. Providing brief examples or explanations will enhance the accessibility of these concepts in the context of psychiatric recovery practices.

Our response (2): Thank you for highlighting this. We have added the following to “Theme: A clinical standardisation of the concept of recovery”, which exemplifies the connection between governmentality and the use of recovery in practice:

“This extract from the text corpus encompassed the twofold aim of risk management and optimisation (governmentality): “The best way to reduce the likelihood of crises from occurring is to develop skills of self-management. These skills create actor awareness, empowerment, and resilience to manage relapse. The ability to recognise and respond to symptoms of mental problems is an important skill. The challenge in relation to recovery is to work with early warning signals in a way that strengthens people's ability to self-regulate instead of creating anxiety about and unnecessary attention to potential relapses.” [24] (Our translation). It accentuated the connection between optimisation (governmentality) and prophylactic practices by potentiating the patients' capabilities of self-monitoring and self-governance.”

3) In line 231, the statement "effectuated a clinical standardization of the concept of recovery" could be further explained to help readers understand its meaning, addressing how clinical concerns and risk management techniques led to standardization and the implications of this standardization.

Our response (3): We have added the following to “The assumptions about recovery”:

”Essentially, a clinical standardisation of the concept of recovery was a manifestation of the conditions of possibilities concerning recovery. For instance, a clinical understanding of and approach to recovery and vice versa mental illness informed the psychiatric conduction of risk management, which dictated the psychiatric response to expressions of risk (based on a clinical interpretation of risk). This could affect patient autonomy negatively and potentially effectuate an increased appliance of coercive measures in psychiatry. In effect, the standardisation of the concept of recovery designated room for manoeuvre for mental health professionals and psychiatric patients in accordance with the limitations and possibilities of standardisation.”

4) More references should be included in the article, especially in lines 45-74, where various types of recovery are mentioned but only one reference is provided for each type.

Our response (4): Thank you for making aware of this need. We have added several references [9,11,12,13,15] to the “Theoretical understandings of recovery” and increased the number of references used in the paper overall.

5) The transformation in personal recovery, discussed in lines 56-61, is unclear and requires further explanation. To facilitate reader comprehension, the time scales for each transformation should be better organized.

Our response (5): We have added this to the “Theoretical understandings of recovery” after careful consideration of your suggestion:

”The transformations involved restructuring of the services provided by a variety of organisations such as psychiatric hospitals (or asylums given the historical context), facilities for persons with handicaps, and prison complexes. Against this backdrop of social transformation and restructuring of established service provisions, the research into recovery conducted in the 1970s and 1980s highlighted the importance of the social aspects of recovery, i.e., recovery was conceived as a personal process within a social context. However, the later conceptions of recovery from the 1990s and onwards have generally disregarded the structural and environmental circumstances as essential for the realisation of personal recovery [7].”

6) In section 1.1, A table could be added to compare the theories.

Our response (6): Thank you for making this suggestion, which we have considered in depth. However, we do not think that the theoretical understandings of recovery lend themselves very well to a table.

Round 2

Reviewer 3 Report

The author made more revisions to this manuscript based on my concerns. I suggest this manuscript should be accepted in the present edition.